

# Characterization of a profilin-like protein from *Fasciola hepatica*

Jessica Wilkie[1,2], Timothy C. Cameron[1,2] and Travis Beddoe[1,2]

[1] Centre for Livestock Interactions with Pathogens (CLiP), La Trobe University, Bundoora, VIC, Australia
[2] Department of Animal, Plant and Soil Science and Centre for AgriBioscience (AgriBio), La Trobe University, Bundoora, VIC, Australia

## ABSTRACT

*Fasciola hepatica* is the causative agent of fasciolosis, an important disease of humans and livestock around the world. There is an urgent requirement for novel treatments for *F. hepatica* due to increasing reports of drug resistance appearing around the world. The outer body covering of *F. hepatica* is referred to as the tegument membrane which is of crucial importance for the modulation of the host response and parasite survival; therefore, tegument proteins may represent novel drug or vaccine targets. Previous studies have identified a profilin-like protein in the tegument of *F. hepatica*. Profilin is a regulatory component of the actin cytoskeleton in all eukaryotic cells, and in some protozoan parasites, profilin has been shown to drive a potent IL-12 response. This study characterized the identified profilin form *F. hepatica* (termed *Fh*Profilin) for the first time. Recombinant expression of *Fh*Profilin resulted in a protein approximately 14 kDa in size which was determined to be dimeric like other profilins isolated from a range of eukaryotic organisms. *Fh*Profilin was shown to bind poly-L-proline (pLp) and sequester actin monomers which is characteristic of the profilin family; however, there was no binding of *Fh*Profilin to phosphatidylinositol lipids. Despite *Fh*Profilin being a component of the tegument, it was shown not to generate an immune response in experimentally infected sheep or cattle.

## INTRODUCTION

Fasciolosis is a worldwide distributed zoonotic infectious disease and constitutes a serious worldwide problem in both humans and livestock (*Mas-Coma, Bargues & Valero, 2018*; *Mas-Coma, Valero & Bargues, 2019*). There are two major pathogens of fasciolosis; *Fasciola hepatica* and *F. gigantica*, which are commonly referred to as liver fluke. The control of fasciolosis has relied upon chemotherapy, predominanantly with the drug triclabendazole. However, due to an over-reliance on this drug in recent years, resistance to triclabendazole has developed (*Fairweather, 2009*; *Kelley et al., 2016*). As triclabendazole is the only drug that will kill both the juvenile and adult life stages of liver flukes, there is an urgent need for the development of novel treatments. For this reason, vaccine development is seen as a sustainable method for the control of *Fasciola* spp. (*Molina-Hernandez et al., 2015*; *Toet, Piedrafita & Spithill, 2014*).

Corresponding author
Travis Beddoe,
t.beddoe@latrobe.edu.au

The current *F. hepatica* vaccine candidates being investigated have shown only moderate protection against *F. hepatica* infection (*Toet, Piedrafita & Spithill, 2014*). The development of an effective vaccine will require a thorough understanding of the host-parasite interactions (*Cwiklinski & Dalton, 2018*; *Cwiklinski et al., 2018*). While the excretory-secretory (ES) products of *F. hepatica* have been thoroughly investigated as vaccine antigens with little sustained success, the tegument surface represents the key interface in host-parasite interactions, performing numerous functions for the parasite such as nutrient absorption, sensory input and protection from the host immune response (*Halton, 2004*). Host antibodies have been demonstrated to have an ability to bind to the tegument antigens of *F. hepatica* (*Howell & Sandeman, 1979*; *Hanna, 1980*; *Sulaiman et al., 2016*; *Cameron et al., 2017*), suggesting that tegument-directed vaccine candidates warrant further investigation. Despite this, there are few reported cases of tegument-directed vaccines. Tegument proteins as vaccine targets in other helminth parasites such as *Schistosoma mansoni* show immense promise, with phase 1 clinical trials in progress (*Fonseca, Oliveira & Alves, 2015*; *Merrifield et al., 2016*; *Molehin, 2020*) and we propose that tegument proteins represent potential novel vaccine antigens for *F. hepatica* control (*Hanna, Anderson & Trudgett, 1988*; *Sobhon et al., 1998*).

The tegument surface of *Fasciola* spp. is a dynamic syncytial layer surrounded by a glycocalyx (*Hanna, 1980*; *Lammas & Duffus, 1983*). Various groups in recent times have attempted to characterize the proteome (*Hacariz, Sayers & Baykal, 2012*; *Ravida et al., 2016*; *Wilson et al., 2011*) and immunoproteome (*Cameron et al., 2017*) of *F. hepatica* using mass spectrometry. The proteome of an enriched tegument extract of *F. hepatica* revealed a range of proteins shared with the schistosome tegument including annexins, tetraspanins, carbonic anhydrase and an orthologue of a host protein (CD59) (*Wilson et al., 2011*). A second study enriched tegument glycoproteins using immobilized lectin chromatography to identify over 369 glycoproteins with a broad range of functions such as proteases, protease inhibitors, paramyosin, venom allergen-like protein II and enolase (*Ravida et al., 2016*). There is an over-abundance of vaccine antigen candidates from the tegument and to narrow down potential candidates, a recent study used a novel ex vivo immunoproteomic technique whereby contact with purified host IgG from infected animals, the flukes will slough (i.e., sheds) its tegument proteins after antibody binding has occurred (*Cameron et al., 2017*). This immunosloughate identified 38 proteins that could be potential vaccine antigens, (*Cameron et al., 2017*). Unsurprisingly, all these tegument proteomic studies identified a large number of cytoskeletal elements such as tubulin, actin and profilin that could be potential vaccine/drug targets due to the crucial function of the *Fasciola* tegument.

Profilins are small actin-binding proteins that are involved in the regulation of actin polymerization by sequestering actin and ADP/ATP exchange (*Krishnan & Moens, 2009*; *Pinto-Costa & Sousa, 2020*). In addition, they are involved in cell signaling between the cell membrane and cytoskeleton by interacting with polyphosphoinositides (PPI) and proline-rich domain containing proteins (*Krishnan & Moens, 2009*; *Pinto-Costa & Sousa, 2020*). In particular, profilin from apicomplexan protozoan parasites such as *Toxoplasma gondii* have been shown to generate a potent IL-12 response in murine DCs activated

through TLR11; as such, profilin from various apicomplexan protozoan parasites have been trialed as vaccine antigens (*D'Angelo et al., 2009*; *Mansilla & Capozzo, 2017*; *Tang et al., 2018*; *Yarovinsky et al., 2005*). Here we describe the identification and biochemical characterization via bioinformatics, phospholipid binding, actin polymerization and poly-L-proline (pLp) affinity that a tegument protein from *F. hepatica* can be classified as belong to the profilin family and we describe its potential use as vaccine candidate.

## MATERIALS AND METHODS

### Cloning and phylogenetic analysis of *Fasciola hepatica* profilin

The native *F. hepatica* profilin sequence (accession number D915_008168) was chemically synthesized and cloned (Bioneer, Oakland, CA, USA) via *NdeI* and *XbaI* sites into a modified pET-28 vector, resulting in an open reading frame containing an N-terminal hexahistidine tag followed by an HRV 3C protease cleavage site and the F. hepatica (*Fh*Profilin) sequence. The *Phylogeny.fr* program was used to compare *Fh*Profilin with identified profilins from other parasitic species and construct a phylogenetic tree based on multiple alignments and a neighbor-joining method as well as to estimate the confidence value of the branching patterns (*Dereeper et al., 2008*).

### Recombinant *Fh*Profilin expression

Plasmid DNA containing the *Fh*Profilin sequence was transformed into BL21 (DE3) *E. coli* cells and plated onto a Luria–Bertani (LB) agar plate containing kanamycin (50 μg ml$^{-1}$). A single colony was inoculated into a starter culture and grown overnight in 10 ml LB medium containing 50 μg ml$^{-1}$ kanamycin with shaking at 225 rpm. The starter culture was used at a 1:100 dilution to inoculate 400 ml of fresh LB medium containing 50 μg ml$^{-1}$ kanamycin and grown at 37 °C until the optical density at wavelength 600 nm (OD600) reached 0.5. Expression of *Fh*Profilin was induced with 0.5 mM IPTG (isopropyl β-D-1-thiogalactopyranoside) and the cells were allowed to grow for a further 4 h at 37 °C. The cells were collected by centrifugation at 6,000$g$ for 10 min at 4 °C and were stored at −20 °C.

### Recombinant *Fh*Profilin purification

The frozen cell pellet was thawed on ice and resuspended in 2 ml of lysis buffer (50 mM NaH$_2$PO$_4$, 300 mM NaCl and 10 mM imidazole at pH 8.0) per gram of wet cell weight. After thawing, 200 μL of 25 mg/ml lysozyme and 200 μL of 2 mg/ml DNase was added and incubated on ice for 30 min. The solution was sonicated with 3 mm microprobe using a Sonics Vibracell VCX 130PB at 25–30% amplitude with 30 s bursts on ice for a total sonication time of 3 min, with 30 s of rest in between each burst. The cell debris was removed by centrifugation at 30,000$g$ for 20 min at 4 °C. The supernatant was added to 1 ml of 50% (w/v) Ni-Sepharose resin (Clontech) pre-washed with 5 column volumes of lysis buffer and incubated for 1 h with gentle shaking at 4 °C. The lysate-nickel Sepharose mixture was loaded into a gravity flow column and the flow through collected. The column was washed with 2 column volumes of wash buffer (50 mM NaH$_2$PO$_4$, 300 mM NaCl and 20 mM imidazole, pH 8.0) and eluted with 8 ml of elution

buffer (50 mM NaH$_2$PO$_4$, 300 mM NaCl and 250 mM Imidazole, pH 8.0) and collected as 2 ml fractions. Each step of the purification process was validated by visualization of protein fractions using SDS–PAGE.

Fractions containing *Fh*Profilin were pooled and concentrated to 5 ml using Amicon ultracentrifugal filters (3 kDa molecular-weight cutoff; Millipore). The partially purified *Fh*Profilin was further purified by size-exclusion chromatography with a Superdex S75 16/60 gel-filtration column (GE Healthcare Life Sciences, Chicago, IL, USA) equilibrated in TBS (10 mM Tris-HCL and 300 mM NaCl, pH 8.0) using an AKTA Basic fast protein liquid-chromatography (FPLC) system at 1 ml/min. The molecular weight, purity and identity of the *Fh*Profilin preparation were confirmed by SDS–PAGE and Western blotting.

### Actin affinity assay

The ability of *Fh*Profilin to bind and sequester actin was investigated. Actin (5 μM) derived from bovine muscle (Sigma, St. Louis, MO, USA) was induced to polymerize with 1 mM MgCl$_2$ and 0.15 M KCl. *Fh*Profilin was added in molar ratios of 1:1, 1:2 and 1:4 molar ratios to the polymerized actin in total volume of 150 μL and incubated at room temperature for 2–3 h. After incubation, the actin-profilin mixtures were centrifuged at 100,000x*g* for 30 min at 20 °C. Equal amounts of the supernatants and pellets fractions were analyzed by SDS-PAGE.

### Phospholipid affinity assay

PIP Strips™ (Echelon Biosciences Incorporated, Salt Lake City, UT, USA) were used to assess the specificity of *Fh*Profilin for associating with various phospholipids. Each membrane has been spotted with 15 assorted phospholipids at 100 pmol in each spot. The membrane was blocked with 5 ml of PBS (50 mM Sodium phosphate, 150 mM NaCl, pH 7.4) plus 3% (w/v) skim milk blocking solution and gently shaken for 1 h at room temperature. The blocking solution was then discarded and various concentrations of *Fh*Profilin, starting with an initial concentration of 20 μg/ml up to 500 μg/ml, was added to 10 ml of PBS plus 3% (w/v) skim milk and gently shaken at room temperature for 1 h.

For the positive control, 5 μg/ml of PI(4,5)P$_2$ Grip™ protein in 5 ml of PBS-T (50 mM Sodium phosphate, 150 mM NaCl, 0.05% (v/v) Tween 20 pH 7.4) plus 3% (w/v) BSA was used. The protein solution was discarded, and the membrane was washed three times with 5 ml of PBS with 5 min of gentle shaking for each wash. After washing the test strip, anti-His HRP-conjugated antibody (R&D systems, Minneapolis, MN, USA) was diluted to 1:10,000 in PBS with 3% (w/v) skim milk for *Fh*Profilin and added to the membrane and incubated for 1 h at room temperature with gentle shaking. For the positive control strip, anti-GST-HRP antibody (GenScript, Piscataway, NJ, USA) was diluted to 1:2,000 in PBS-T 3% (w/v) BSA. The antibody solution was discarded, and the membrane washed as previously stated. For both *Fh*Profilin and positive control samples, detection was performed incubating the mebrane with 5 ml of Clarity ECL substrate (Bio-Rad Laboratories, Hercules, CA, USA) for 5 min and imaged using the C-DiGiT blot scanner (Li-Cor).

## Poly-L-proline affinity assay

The affinity of *Fh*Profilin for proline-rich domains was investigated. pLp sepharose was prepared by coupling 50 mg of pLp (Sigma-Aldrich, St. Louis, MO, USA) to 1 g of cyanogen bromide-activated sepharose resin (Sigma-Aldrich, St. Louis, MO, USA). The pLp was dissolved in 4 ml of ice cold deionized water and the sepharose was resuspended in 8 ml of 250 mM sodium carbonate to make a 50% slurry. The pLp was added to the 50% slurry and stirred for 2 h at room temperature. The mixture was transferred to a cold room and stirred at 4 °C overnight. The reaction was quenched with 1.2 ml of 10× bead buffer (1 M NaCl, 1 M glycine and 100 mM Tris). The resin was washed in a Buchner funnel with 500 ml of deionized water, dried and stored at 4 °C in 1× storage buffer (10 mM Tris at pH 7.5, 50 mM KCl, 1 mM EDTA and 0.002% (w/v) sodium azide). A total of 50 μL aliquots of the 50% pLp sepharose slurry with PBS was incubated with 10 nM of purified *Fh*Profilin for test samples and 10 nM of BSA for control samples. Each sample was incubated at room temperature for 15 min and analyzed by SDS PAGE. To quantify the binding effect of *Fh*Profilin with pLp, different concentrations of *Fh*Profilin, from 1 to 10 μg/μL, were incubated with the poly-L-proline sepharose beads for 15 min and analyzed by SDS-PAGE.

## Immune profile of *Fh*Profilin in sheep and cattle sera

*Fasciola hepatica* were obtained from the abattoir and whole worm extract (WE) was prepared as previously reported (*Swan et al., 2019*). Native GST was purified as previously described by (*Wijffels et al., 1992*). A total of 40 μg of WE, purified *Fh*Profilin (3 μg) and native GST (3 μg) were loaded onto an SDS-PAGE and transferred to PVDF membrane (Bio-Rad Laboratories, Hercules, CA, USA) using a Trans-Blot® Turbo™ Transfer System (Bio-Rad Laboratories, Hercules, CA, USA). After blocking with 5% (w/v) skim milk, pooled sera from experimentally infected Merino sheep (12 animals pooled, 6 weeks post-infection, infected with 200 metacercariae), Indonesian Thin Tailed (ITT) sheep (15 animals pooled, 6 weeks post-infection, infected with 200 metacercariae) and cattle (six animals pooled from 125 days post-infection, infected with 350 metacercariae) (kindly donated by Prof. Terry Spithill) were incubated at a 1:4,000 dilution with the blots in 5% (w/v) skim milk with PBS-T for 1 h. Membranes were washed three times with PBS-T and were then incubated with either anti-sheep HRP-conjugated IgG or anti-bovine HRP-conjugated IgG (Sigma-Aldrich, St. Louis, MO, USA) diluted to 1:4,000 in 5% (w/v) skim milk in PBS-T for 1 h. Membranes were washed again as above and thens were visualized by Clarity ECL substrate (Bio-Rad Laboratories, Hercules, CA, USA) and C-DiGiT blot scanner (Li-Cor) according to the manufacturer's instructions.

## RESULTS AND DISCUSSION

### Bioinformatic analysis of the *Fasciola hepatica* profilin gene

The profilin gene identified (accession number D915_008168) in *F. hepatica* (*Fh*Profilin) has an open reading frame (ORF) of 393 bp, encoding a 130 amino acid protein with a predicted molecular mass of 14 kDa and a predicted isoelectric point of 5.35, which was predicted using the "ProtParam" tool from ExPASy Bioinformatics Resource Portal

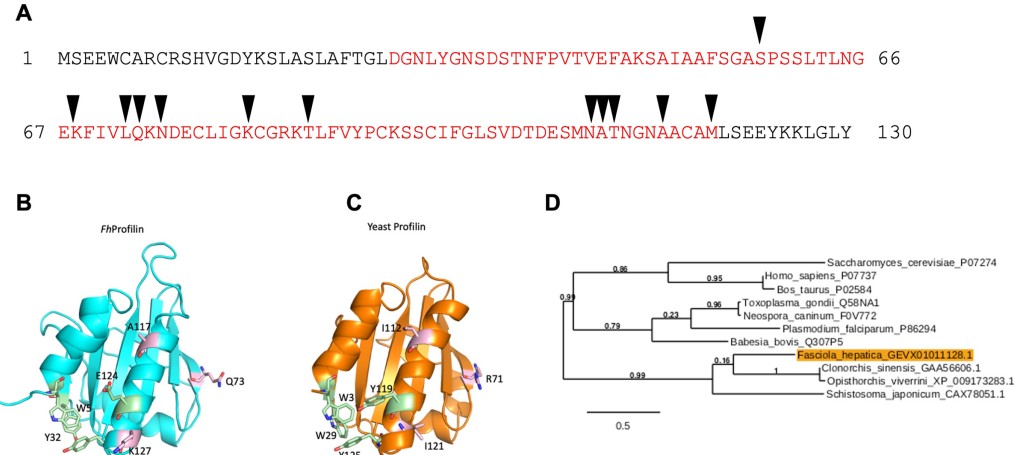

**Figure 1 Bioinformatic analysis of *F. hepatica* profilin (*Fh*Profilin) gene.** (A) The amino acid sequence for *Fh*Profilin (accession number D915_008168). The red letters indicate the profilin family domain. Black triangles indicate the putative actin-binding sites. (B and C) Homology model of *Fh*Profilin showed as a ribbon (B) and the crystal structure of *Saccharomyces cerevisiae* profilin (1YPR) (C). The residues involved in phosphatidylinositol phosphate interaction (light pink) and proline binding (light green). (D) Phylogenetic tree of the *F. hepatica* profilin was produced online using *Phylogeny.fr* (*Dereeper et al., 2008*). The genebank or UniProtKB accession numbers used to construct the tree appear after each species name. Numbers shown at branch nodes indicate bootstrap values.

(https://web.expasy.org/protparam/). A domain search using InterPro (http://www.ebi.ac.uk/interpro/), revealed the presence of a conserved profilin domain, containing putative actin binding sites (Fig. 1A). Structural homology modeling revealed that *Fh*Profilin is structural indentical to *Saccharomyces cerevisiae* profilin with conservation of residues involved in proline binding however residues in a putative phosphatidylinositol 4,5-bisphosphate (PIP2)-interaction site are not conserved (Fig. 1B; Fig. S1).

In a phylogenetic analysis of profilin proteins from other parasite species, it was revealed that *Fh*Profilin clustered in a clade alongside other trematode species with identities ranging from 31.1% (*Schistosoma japonicum*) to 41.5% (*Clonorchis sinensis*), while profilins from apicomplexan parasites clustered in a separate clade (Fig. 2).

## Protein expression and purification of *Fh*Profilin

*F. hepatica* was recombinantly expressed in *E. coli* and purified via immobilized metal-ion affinity chromatography, and when visualized using SDS-PAGE resulted in a single protein with an expected molecular mass of 14 kDa showing that *Fh*Profilin was successfully expressed and purified as a soluble protein (Fig. 2A). To further purify and characterize *Fh*Profilin it was subjected to size-exclusion chromatography. A typical trace (Fig. 2B) revealed that *Fh*Profilin elutes at the volume of approximately 75 ml which corresponds to a molecular weight of approximately 30 kDa, suggesting that *Fh*Profilin is a dimer in solution. Several previous studies have reported that profilin from different species such as yeast, birch pollen and humans can form dimers and tetramers in solution (*Babich et al., 1996*; *Mittermann et al., 1998*; *Wopfner et al., 2002*). The oligomeric state of profilin is important for the function of certain profilins, for example, the allergenic potential of birch

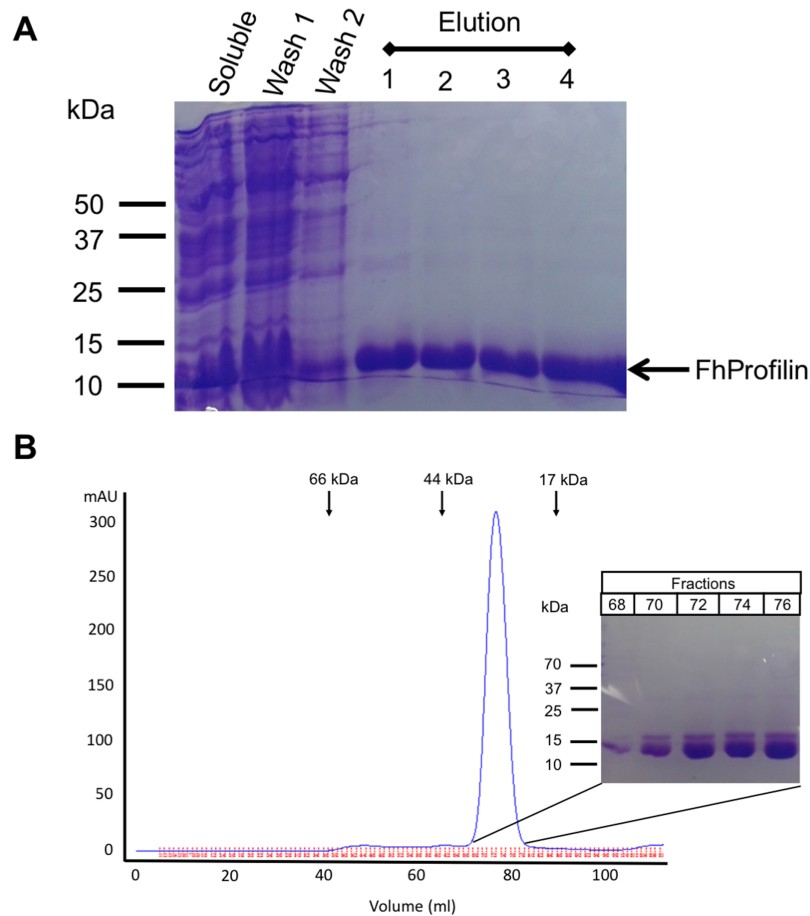

**Figure 2 Recombinant expression and purification of *Fh*Profilin.** (A) Cell lysate expressing *Fh*Profilin were applied to a NI-IDA column and washed twice before elution with imidazole. A total of 15 µL of each stage of the purification was resolved by SDS-PAGE and stained with Coomassie blue. (B) Size-exclusion chromatography trace of recombinant *Fh*Profilin. Arrows indicate the elution volumes of proteins of known molecular weight. Insert: 15 µL of each fraction was resolved by SDS-PAGE and stained with Coomassie blue.

pollen is higher if profilin is in a dimeric state (*Mares-Mejia et al., 2016*). In addition, interactions with phosphoinositides and pLp is regulated by an oligomeric form of profilin, with dimers having a weaker affinity to pLp than tetrameric profilin (*Korupolu et al., 2009*), suggesting the oligomeric state of *Fh*Profilin may regulate its function.

## Biochemical characterization of *Fh*Profilin

Profilins are characterized by having three major biochemical functions (*Krishnan & Moens, 2009*; *Moreau et al., 2017*, *2020*); firstly, profilins bind to and sequester actin monomers, therefore affecting how actin filaments polymerize. Secondly, profilins have an affinity for proline-rich domains contained within their interacting ligands and lastly, they have an affinity for phosphatidylinositol lipids that dissociate the actin:profilin complexes.

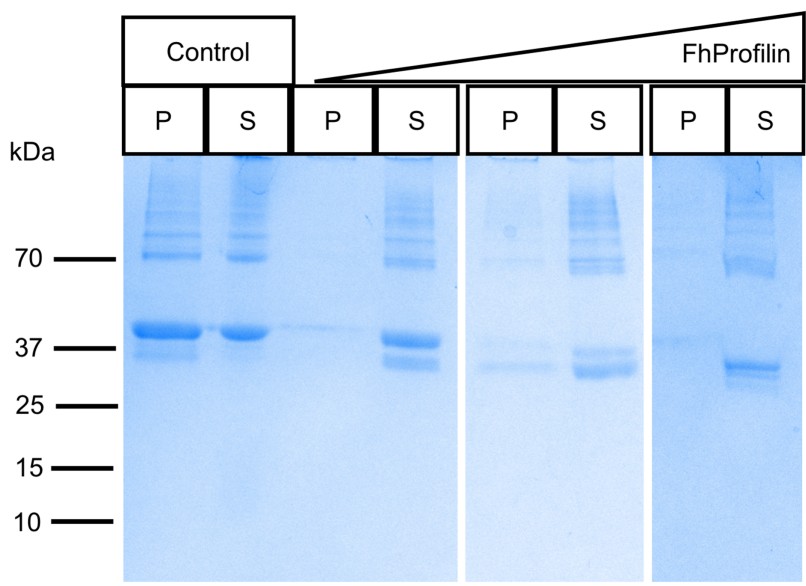

**Figure 3 SDS-PAGE analysis of polymerized actin incubated with *Fh*Profilin.** Different ratios of actin to *Fh*Profilin (1:1, 1:2 and 1:4) were incubated and separated into polymerized or monomeric actin fractions by centrifugation. Control lanes contained no *Fh*Profilin. The pellet (P) and supernatant (S) fractions (15 μL) were resolved by SDS-PAGE and stained with Coomassie blue.

To determine if *Fh*Profilin is able to perform the classic functional profile of other profilins, several in vitro assays were performed. Recombinant *Fh*Profilin was added to polymerizing bovine actin at different ratios and monomeric actin was separated from polymerized actin by centrifugation. At all ratios of actin:profilin, the majority of actin appeared in the soluble supernatant fraction with very little polymerized actin being observed in the pellet fractions (Fig. 3). This indicates that *Fh*Profilin has a strong ability to sequester actin similar to other profilins from yeast and humans (*Eads et al., 1998*; *Pinto-Costa & Sousa, 2020*). A major defense mechanism of *F. hepatica* is the ability to shed its tegument proteins after binding by antibodies, which is highly dependent on cytoskeletal rearrangement, thus making *Fh*Profilin an ideal vaccine candidate or drug target (*Hanna, 1980*). Human and *Plasmodium* profilin have been explored as possible drug targets and many of the major drugs against *Fasciola* target tubulin, thus cytoskeletal components present ideal drug targets (*Fairweather et al., 2020*; *Kumpula & Kursula, 2015*; *Moens & Coumans, 2015*). In the future, the actin-profilin interaction could be explored as possible drug target in *Fasciola*.

Profilin interacts with its many ligands via proline-rich sequences (*Bjorkegren et al., 1993*; *Kursula et al., 2008b*; *Mahoney, Janmey & Almo, 1997*). The majority of residues involved in polyproline binding are conserved in *Fh*Profilin suggesting it interacts with a variety of partner proteins (Fig. 1A). To assess the ability of *Fh*Profilin to interact with polyproline, it was subjected to pulldown assays using polyproline-sepharose beads (Fig. 4). *Fh*Profilin was successfully detected in the bound fraction of the assay, whereas

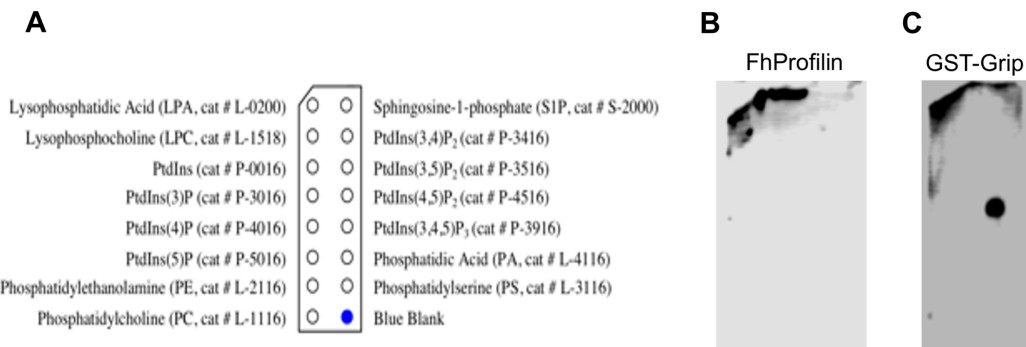

**Figure 4 Binding of _Fh_Profilin to phospholipids.** (A) The location of the different phospholipids on the membrane (Echelon Biosciences Incorporated, Salt Lake City, UT, USA). (B) _Fh_Profilin and (C) GST-Grip respectively binding to phospholipid microarrays detected using an anti-hexhistidine and anti-GST antibody.

BSA was not (Fig. 4), showing that the interaction with polyproline-sepharose is specific to _Fh_Profilin. Not all _Fh_Profilin was bound to the resin, suggesting either the interaction was weak, or capacity of the resin was exceeded (Fig. 4). It appears that _Fh_Profilin may bind to similar physiological substrates as human profilin and therefore is likely to be involved in other cellular functions such as ribonucleoparticle processing (_Giesemann et al., 1999_), mRNA splicing (_Skare et al., 2003_) and nuclear export (_Stuven, Hartmann & Gorlich, 2003_).

The ability of profilins to bind phosphatidylinositol lipids is important as it can regulate phosphoinositide metabolism and its ability to move from the membrane to the cytosol where it can interact with actin or other ligands (_Chaudhary et al., 1998_; _Lambrechts et al., 2002_). The phosphatidylinositol lipid specificity of _Fh_Profilin was assessed using a mini phosphatidylinositol lipid array (Fig. 5). There was no detectable phosphatidylinositol lipid binding by _Fh_Profilin even at the highest concentration of 500 μg/ml (Fig. 5B). However, the positive control protein Grip supplied with the array was positively identified binding to the appropriate lipid phosphatidylinositol 4, 5-bisphosphate (Fig. 5C) confirming the validity of the assay and suggesting that _Fh_Profilin has very weak or no association with phosphatidylinositol lipids. It is not surprising that _Fh_Profilin does not bind phosphatidylinositol lipids as only two out of five PIP-2 binding sites are conserved (Fig. S1), which are normally seen in other members of the profilin family (_Munkhjargal et al., 2016_). Human profilin has been shown to bind to PI(3,4)P$_2$, PI(3,4)P$_2$ and PI(3,4,5)P$_3$ which is silimar to bovine profilin that binds PI(3,4,5)P$_3$ and PI(4,5)P$_2$ with some binding to PI(3)P, PI(4)P, and PI(5)P (_Kursula et al., 2008a_; _Lu et al., 1996_). This different which has been observed between the more closely related profilins from the apicomplexan parasites _Plasmodium_ and _T. gondii_ that _Plasmodium_ profilin can bind phosphatidylinositol lipids (PI(4)P, and PI(5)P) whereas _T. gondii_ profilin cannot (_Kucera et al., 2010_; _Kursula et al., 2008a_). The lack of binding of _Fh_Profilin to phosphatidylinositol lipids may suggest the diffenence in phosphatidylinositol lipid metabolism in _F. hepatica_.

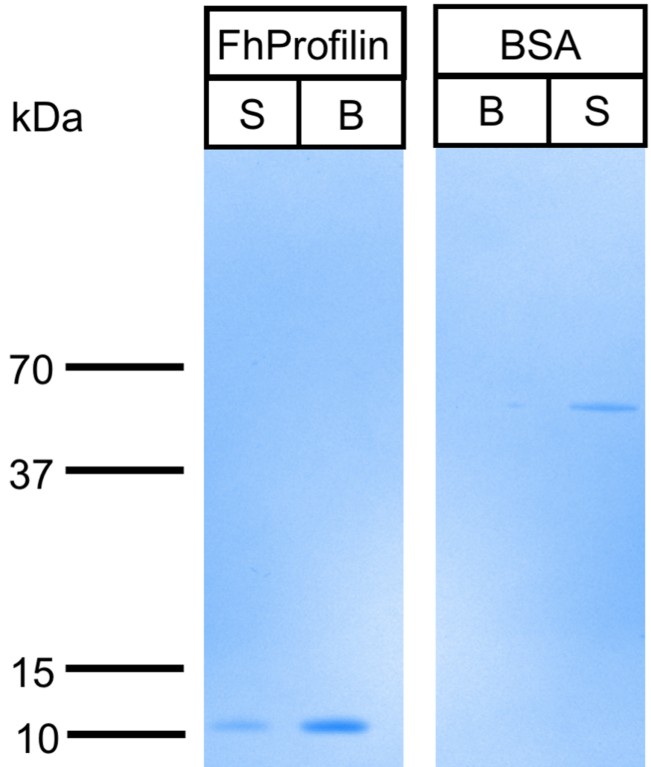

**Figure 5 Binding of *Fh*Profilin to polyproline sepharose.** Equal amounts of *Fh*Profilin and BSA were passed over polyproline sepharose. Unbound (S) and Bound (B) proteins were visualized by SDS-PAGE and Coomassie staining.

## Immune profile in cattle and sheep

Apicomplexan parasitic profilins have the ability to stimulate the immune system via toll-like receptor 11 (TLR11) due to the presence of a parasite-specific surface motif consisting of an acidic loop followed by a long β-hairpin insert (Fig. S1) (*Kucera et al., 2010*). Due to this immune modulation activity, profilins have been used as potential vaccine candidates and adjuvants, in particular against *T. gondii* and *Eimeria* spp. (*Jang et al., 2011a*, *2011b*, *2011c*; *Tanaka et al., 2014*). To ascertain if *Fh*Profilin was exposed to the host immune system, pooled sera from experimentally *Fasciola* infected animals was tested via Western Blot (Fig. 6). Immune sera from infected animals consisting of the susceptible sheep breed (Merino), the *F. gigantica*-resistant Indonesian Thin Tail (ITT) sheep and cattle did not recognize the recombinant *Fh*Profilin while native *F. hepatica* glutathione S-transferase (n*Fh*GST), a major parasite excretory-secretory antigen, was recognized (*LaCourse et al., 2012*). This suggests that *Fasciola* profilin is not exposed to immune system during infection (Fig. 6). The lack of *Fh*Profilin-specific antibodies from exposed animals is unexpected as profilin from other parasites such *Babesia* spp. (*Munkhjargal et al., 2016*) and *Schistosoma japonicum* do elicit an immune response post infection (*Zhang et al., 2008*). Further the use of electron microscopy to perform ultrastructural studies to determine where within the tegument *Fh*Profilin is located and whether this protein is exposed to the host immune system would help to assess the

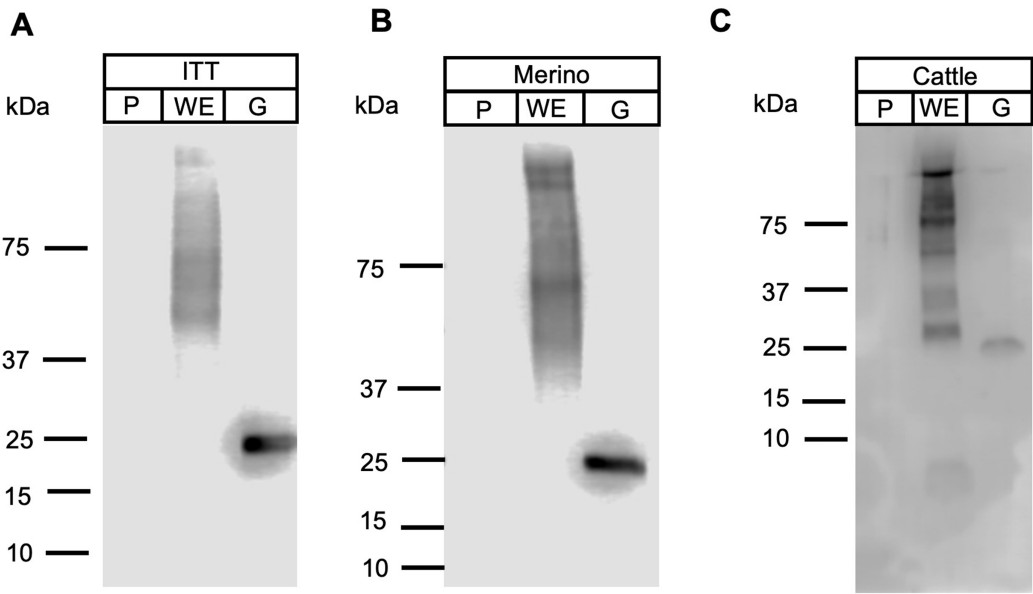

**Figure 6 Western blot analysis of *Fh*Profilin with immune sera.** Immune sera from infected ITT sheep (A), Merino sheep (B) and cattle (C) was probed on whole fluke extract (WE), recombinant *Fh*Profilin (P) and native GST (G). The experiment was repeated at least three times.

validity of *Fh*Profilin as a vaccine candidate. A lack of direct exposure on the tegument should not exclude *Fh*Profilin as a vaccine candidate, as the "hidden" antigen vaccines are against the nematode *Haemonchus contortus* (*LeJambre, Windon & Smith, 2008*; *Munn, 1997*) and the tick *Rhiphicephalus microplus* (*Willadsen & Kemp, 1988*) demonstrate the commercial viability of targeting antigens of this nature, as long they are essential in function to the parasite. Profilin is essential for the survival of *P. falciparum* (*Kursula et al., 2008a*) and necessary for virulence of *T. gondii* (*Plattner et al., 2008*), suggesting that profilin could also be essential for the survival and pathogenesis of *F. hepatica*.

## CONCLUSIONS

*Fasciola* is a zoonotic infection of worldwide concern which until recently was successfully controlled through the use of triclabendazole; however, the appearance of drug resistant parasites has required the need for the development of a vaccine or new drug targets. We have characterized a putative open reading frame that has homology to the profilin family. Profilin plays an essential role in regulating the actin cytoskeleton in all eukaryotic cells. The recombinant *Fh*Profilin displayed hallmark biochemical features of other profilins by binding to actin and polyproline, however the lack of binding to phosphatidylinositol lipids suggests that phosphatidylinositol lipid metabolism may be different in *Fasciola* compared to other parasite species. Despite recombinant *Fh*Profilin not being recognized by immune sera from infected animals, the use of *Fh*Profilin as a potential vaccine candidate is worth further investigation due to the predicted critical function of this protein to the parasite's pathogenesis and survivability.

## ACKNOWLEDGEMENTS

We thank Jaclyn Swan for help with analyzing the *Fasciola hepatica* genome database, Gemma Zerna for the native GST protein and Prof. Terry Spithill for the donation of sheep and cattle immune sera.

### Funding

The authors received no funding for this work.

### Competing Interests

The authors declare that they have no competing interests.

### Author Contributions

- Jessica Wilkie conceived and designed the experiments, performed the experiments, analyzed the data, prepared figures and/or tables, authored or reviewed drafts of the paper, and approved the final draft.
- Timothy C. Cameron performed the experiments, analyzed the data, authored or reviewed drafts of the paper, and approved the final draft.
- Travis Beddoe conceived and designed the experiments, analyzed the data, prepared figures and/or tables, authored or reviewed drafts of the paper, and approved the final draft.

### Data Availability

Raw Data is available at Open at La Trobe: Beddoe, Travis; Cameron, Tim (2020): *Fh*Profilin data.zip. La Trobe. Figure. DOI 10.26181/5ed48514efce1.

### Supplemental Information

Supplemental information for this article can be found online at http://dx.doi.org/10.7717/peerj.10503#supplemental-information.

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
