# Peer review of "Characterization of a profilin-like protein from Fasciola hepatica"

_PeerJ, doi:10.7717/peerj.10503_

## Round 0.1 · original submission · Major Revisions

All reviewers including myself agree the manuscript deserves to be published. Although there is considerable merit in your paper, we also identified some concerns that must be considered in your resubmission. All three reviewers have contributed significant feedback, with detailed analysis and valuable comments on how to improve the manuscript. As you address the comments, please pay particular attention to their suggested improvements.

Reviewer 1 ·

Basic reporting

No Comment

Experimental design

No Comment

Validity of the findings

No Comment

Additional comments

This is a sound and interesting study.
I have some suggestions for the authors
First, they should consider making molecular models of the protein. This is relatively straight forward using online servers such as Phyre2. This may help explain some of the results and differences between the fluke protein and ones from other species.

Second, they should consider measuring ATPase activity and comparing this to other species.

Reviewer 2 ·

Basic reporting

Wilkie, Cameron and Beddoe present the successful production of F. hepatica profilin in bacteria, purified using an N-terminal His-tag followed by size-exclusion chromatography. This protein is present in the tegument and thus thought to be important for parasite-host interaction. An protease site was present between the His-tag and profilin sequence, but not utilised to remove the tag.
The authors go on to show that the produced profilin binds to actin and keeps actin monomers soluble, preventing them from polymerising. Other profilins also do this. They also show the profilin binds to polyproline resin, does not bind the PIP variants they tried and is not recognised by Fasciola hepatica susceptible sheep or cattle immune serum.
In general, the manuscript is well-written and intelligible. However, the sentence on lines 245-247 reads a bit clunky in my opinion.
The manuscript does give the impression that all experiments were only performed once and not optimised (a bit "quick-and-dirty", although this sounds a bit too negative).
- line 285. the parasite-specific surface motif consisting of an acidic loop followed by a long beta-hairpin insert could be indicated in Figure 1. In that respect, another useful addition to Figure one would be a structural model, with the polyproline and PIP binding residues hightlighted. It should be possible to generate a useful structural model by homology modelling, even using a webserver.
- please provide full gels and blots for all figures as supplementary information, i.e. raw data.

Experimental design

- sonicated at 25-30% amplitude doesn't say very much without detailing the exact equipment and probe used.
- the polyproline resin binding experiment would be more informative with a range of profilin / resin concentrations, including proportions where the resin is limiting and where the profilin is limiting. In that way, it could be distinguished if a fraction of the profilin sample is unable to bind resin or if there was just not enough resin to bind all the protein.
- how likely is it that in the PIP mini-array the PIP(s) that profilin might bind are just not there? In this respect a control with a profilin known to bind PIPs would have been welcome. Or at least a summary on which profilins bind exactly which PIPs (and if these PIPs are in the array). In any case, the molecular identities of the 15 PIPs used in the micro-array need to be stated.
- Figure 6 is rather "ugly"

Validity of the findings

- replication of some of the experiments would have been welcome; or a statement as to how many times experiments were performed with similar results.
- after size-exclusion chromatography, the gel appears to show a double band. Was this analysed by mass-spec of N-terminal sequence analysis? Perhaps this was the result of spontaneous cleavage of the tag?
- please provide full gels and blots for all figures as supplementary information, i.e. raw data.
- conclusions are reasonable and unwarranted speculation is absent.

Additional comments

I could not locate the database ID quoted (accession number GEVX010111281, which database?), but the protein sequence appears to be the same as that of GenBank entries THD21072.1 (hypothetical protein D915_008168 [Fasciola hepatica]) and TPP67514.1 (hypothetical protein FGIG_07142 [Fasciola gigantica]). I think it would be good to add this information and/or where the original database ID can be found.
The last three amino acids in the sequence shown in Fig. 1 are GYL, while the above-mentioned GenBank entries have GLY. Could this be a typo?
Figure 5 is discussed before figure 4, so they would be better swapped.

·

Basic reporting

Please see the general comment section.

Experimental design

Please see the general comment section.

Validity of the findings

Please see the general comment section.

Additional comments

The submitted manuscript, Wilkie et al, is an interesting read in which the authors have expressed, purified and characterized a profilin like protein in Fasciola hepatica. The paper is well organized and describes the introduction, the observations, and the methodology in explicit details. The results obtained are also reasonable. This paper represents an important contribution to the infectious disease community. However, there are major flaws with result interpretations that need to be revised before the manuscript can be considered for resubmission. My comments are given below:
Major concerns:
• Please summarize the key findings and associated methods in brief in the introduction.
• The authors claim that the immunized sera showed no reactivity to FhProfilin. However, the western blot experiment (fig 6) lacks a positive control showing that the immunized sera can detect any protein (at all). Please comment on it.
• Phospholipid affinity assay: The authors claim that unlike conventional profilins, FhProfilin does not bind to phospholipids. To test their hypothesis, the authors performed Phospholipid affinity assay. To further support their findings, they incorporate a positive control that ,unlike the FhProfilin, binds to phospholipids.
However, there is a major flaw with the experimental design. The buffers used for FhProfilin and control protein are not identical. The buffer used for the control protein contains additional tween 20. I suspect it may affect the experiment.
• Immunization of animals: The manuscript lacks proper details of immunization of animals. For example, age, sex, etc. How many times the animals were immunized, days interval, what dose? Etc.
• I highly recommend that the authors should include multiple sequence alignment file in the supporting information to show the conserved lipid binding and polypeptide binding site residues in different profilins across organisms. Currently the authors claim that FhProfilin contains 2 out of 5 PIP2 binding residues. Please show the residues in multiple sequence alignment figure.

Other minor issues:

Introduction
• Please define the term ‘tegument membrane’ at its first occurrence in the manuscript.
• In ‘FhProfilin’ Fh represents an organism and thus should be italicized.
• Line 77: ‘….potential vaccine/drug targets due to their essential function of the Fasciola tegument….’
Please incorporate reference to show that FhProfilin is essential for the pathogen survival. If not available, please consider rephrasing the sentence.

Materials and methods:
• Line 103: ‘50 mg ml-1’
I believe it is a typo. If yes, please replace mg/ml with ug/ml
• Line 107: Please elaborate the term OD600
• Line 119: I assume the Ni-sepharose resin was pre-washed with the loading buffer before adding the supernatant. If yes, please incorporate the details.
• Line 138: ‘FhProfilin (actin/FhProfilin)’
o What does actin/FhProfilin represent here.
o Please include the details of final volume of each reaction.
• Line 163: ‘..according to the manufacturer’s instructions.’
Please provide a brief detail of the method.
Results and Discussion:
• Line 199: ‘(accession number GEVX010111281)’
Is it uniprot id? Gene id? Please provide details.
• Line 231: Please include references about recent studies on Plasmodium profilin. (PMIDs: 32034083 and 28552953)
• Line 296: Please consider replacing ultrastructural with structural.
Figures and table
• Fig 1A: Please highlight Fh in the tree
• Fig 2B: Please replace ‘Fractions’ with ‘Fractions (ml)’
• Fig 3: All the lanes in the SDS-PAGE image are smeary. Please comment on it.
• Fig 5: If possible, please replace the current image with a single gel image showing all the lanes and the marker lane.
• Fig 6: The gel lacks a positive control showing the immunized sera can recognize anything at all.

---

## Round 0.2 · Minor Revisions

The authors addressed the main concerns of the reviews. However, the revised manuscript still deserves attention. Please, improve the quality of Figures 6A and 6B: I agree with Reviewer #1 that the figure is too adjusted to show the main results. In my opinion, it is advisable to provide the original photos of the Western Blot (authors mentioned the experiments were repeated three times) to avoid any doubt related to the FhProfilin recognition by pooled sera from experimentally Fasciola infected animals.

Reviewer 2 ·

Basic reporting

Figures 6A and 6B are still unconvincing, low-res and they look like the contrast has been manipulated too much. I would recommend that somebody with more knowledge of Western blots looks at this, and at the raw data the authors may provide, so that the reliability of this information can be judged.

Experimental design

No further comment.

Validity of the findings

No comment.

Additional comments

The authors made a consistent effort to address all the reviewer concerns, although an important part of the additional experiments/controls were not performed.

·

Basic reporting

The submitted manuscript, Wilkie et al, is an interesting read in which the authors have expressed, purified and characterized a profilin like protein in Fasciola hepatica. The paper is well organized and describes the introduction, the observations, and the methodology in explicit details. The results obtained are also reasonable. This paper represents an important contribution to the infectious disease community. However, there is major flaws with result interpretations that needs to be revise before the manuscript can be considered for resubmission. My comments are given below:

Experimental design

The paper is well organized and describes the introduction, the observations, and the methodology in explicit details. The results obtained are also reasonable.

Validity of the findings

The paper is well organized and describes the introduction, the observations, and the methodology in explicit details. The results obtained are also reasonable.

---

## Round 0.3 · accepted · Accept

The authors addressed the main concerns from the reviewer.